# Effect of Replacement of Synthetic vs. Natural Curing Agents on Quality Characteristics of Cinta Senese Frankfurter-Type Sausage

**DOI:** 10.3390/ani10010014

**Published:** 2019-12-19

**Authors:** Silvia Parrini, Francesco Sirtori, Anna Acciaioli, Valentina Becciolini, Alessandro Crovetti, Oreste Franci, Annalisa Romani, Arianna Scardigli, Riccardo Bozzi

**Affiliations:** 1Department of Agricultural, Environmental, Food and Forestry Science and Technology, University of Florence, Via delle Cascine 5, 50144 Firenze, Italy; francesco.sirtori@unifi.it (F.S.); anna.acciaioli@unifi.it (A.A.); valentina.becciolini@unifi.it (V.B.); alessandro.crovetti@gmail.com (A.C.); oreste.franci@unifi.it (O.F.); riccardo.bozzi@unifi.it (R.B.); 2Pharmaceutical, Cosmetic, Food Supplement, Technology and Analysis, Department of Statistics Computer Science Applications “G. Parenti”, University of Florence, Via U. Schiff, 6, 50019 Sesto Fiorentino, Firenze, Italy; annalisa.romani@unifi.it (A.R.); ari.scardigli@gmail.com (A.S.)

**Keywords:** natural extract, curing agents, sausages, Cinta Senese pig

## Abstract

**Simple Summary:**

The increasing demand of natural and environmentally sustainable foods promotes the use of natural extract as curing agents in meat products. In this context, the extensive rearing system, characteristic of autochthonous pigs, could represent an added value for the consumers. Moreover, the introduction of new types of products to enhance the second-choice portion of meat can be economically important. The present study aims to test and to evaluate the quality traits during storage times of frankfurter-type sausages of Cinta Senese using natural extract to replace nitrites and nitrates as curing agents. Results of this study demonstrate that those products are safe for the human consumption, but some sensorial and physical traits could be improved. The implementation of a new product using meat of a local breed and agricultural by products as curing agents could be interesting both in terms of sustainability and valorization of the territory and of Cinta Senese products.

**Abstract:**

Frankfurter-type sausages (called sausages) were manufactured using Cinta Senese meat. Two different formulations were considered: (i) nitrite and nitrate as curing agents (NIT), (ii) natural mixture (NAT) totally replacing the synthetic curing agents. Microbiological, chemical, and physical characteristics during three different storage times (7, 30, 60 days) were investigated, while sensorial traits were evaluated at the end of the period. The main foodborne pathogens (*Escherichia coli*, *Listeria monocytogenes*, coagulase positive *Staphylococcus* spp., *Salmonella* spp., total bacterium at 30 °C) were absent in both sausage groups. Both types of sausage had a high content of fat probably due to the high intramuscular fat of the local breed. The fatty acid composition of NAT sausages would seem slightly less efficient in the lipid oxidation control. Regarding color parameters, NIT sausages showed greater lightness and redness, while NAT ones were more yellow, thanks to the effect of nitrate on color. All texture parameters resulted higher in NIT, except for the springiness. Storage time mainly affected total microbial count, pH, and color. The addition of natural extract changed the perception of some sensorial properties above all in terms of taste and odor. Natural extract represented an alternative to synthetic additives in Cinta Senese sausages even if some attributes could be improved.

## 1. Introduction

Cinta Senese is an Italian local pig breed characterized by the production of high-quality meat with specific regional identity, traditionally processed into pork products in order to allow the extension of shelf-life respect to the fresh meat. Among them, Frankfurter-type sausage does not represent a typical cured meat product but may be an opportunity of innovation in the specific framework and an improvement in the use of second-choice portion of meat. In Tuscany, Cinta Senese is often reared in marginal areas where farms usually support the animal rearing with other productions such as olives in hilly areas and chestnuts in mountain areas in order to have more sources of income. These types of agricultural systems are accompanied by the production of byproducts that could be used in the food sector to form additives in curing products, increasing the link between the breed and its origin.

Traditionally, curing agents included in processed meat products are sodium and potassium salts of nitrite and nitrate additives and their use is authorized by European Union within a safe level [1]. Their role includes the capacity to inhibit the growth of pathogenic bacteria and to delay oxidative rancidity [2], leading to high stability and increasing the product shelf-life. These curing agents are also involved in the stabilization of the sensory properties [3] and have a positive effect on color, cured cooked flavor, and aroma [4]. However, the nitrite-nitrate consumption may represent a risk for human health. Ingested sodium and potassium salts of nitrite and nitrate are in the major part excreted as nitrate except for a small portion that recirculates through salivary glands, and it is converted by mouth bacteria into nitrite [1]. The absorption of an excessive amount of nitrite could cause an oxidation of hemoglobin into methemoglobin leading a reduction of “the ability of red blood cells to bind and transport oxygen through the body” [1]. Nitrite in food (and nitrate converted to nitrite in the body) may also contribute to the formation of a group of compounds known as nitrosamines (also called *N*-nitrosamines) some of which are associated to the formation of carcinogenic and mutagenic compounds [5]. To the best of our knowledge, besides to the amount of nitrite added to products, a wide range of factors may potentially affect the formation of nitrosamine such as meat quality and its fat content, processing condition as heat applied during drying or smoking, packaging, and storage/maturation conditions [6,7].

The increasing demand of consumers for natural foods moved the food industry interest to include natural curing agents in foods and in processing foods [8]. The use of natural extracts as antioxidants and antimicrobials has advantages such as high consumer acceptance and healthiness, although some studies [8,9] reported disadvantages such as their higher cost and lower effectiveness.

The total antioxidant capacity of fruit and vegetable extracts is based on the concentration of some compounds, such as ascorbic acid, alpha-tocopherol, beta-carotene, flavonoids, and phenols. Furthermore, some authors [10,11] have highlighted that the antimicrobial and antioxidant properties were primarily associated to phenolic compounds content, including volatile compounds. The antioxidant activity of phenolic compounds is given by three mechanisms: free-radical scavenging activity, [12] transition-metal-chelating activity, and/or singlet-oxygen quenching capacity [12,13]. The antimicrobial activity is enabled thanks to the presence of hydroxyl groups and their relative position in the phenolic ring [8].

Some studies [14,15] proposed the addition of natural compounds to the traditional recipe, that foresees synthetic additive, in order to enhance the qualitative properties of products. Ayo et al. [16] studied the effect of replacement of pork backfat with vegetable fat on frankfurter sausages indicating a health improvement of nutritional profile with a 25% addition of walnuts. The main concern of the various studies remains to find an alternative to synthetic curing agents able to address the multiple activities covered by the latter considering that there is no current single substitute for such compounds, particularly for nitrite and nitrate. Sebranek et al. [17] and Eskandari et al. [14] reported successful nitrite/nitrate additive reduction in processed meat products. Alirezalu et al. [18] working on the substitution of curing agents in commercial frankfurter-type sausages of unspecified pork with natural antimicrobial compounds (nisin, ɛ-polylysine, or chitosan) and antioxidant mixture suggested further research to improve color stability, storage, and other physical properties. Moreover, as highlighted in a review of Velasco et al. [8], some experimental studies have been focused on factors affecting natural extracts’ activity such as the specific extraction process or the measurement method. The extraction of natural extract could follow a green chemistry procedure using a sustainable and eco-friendly methodology as the case reported by Romani et al. [19]. In this context the use of natural antioxidant/antimicrobial obtained from agricultural byproducts which otherwise would be wasted could become a good alternative.

The use of grape seeds (*Vitis vinifera*), chestnut (*Castanea sativa*), and olive oil (*Olea europaea*) characterized by antioxidant and antimicrobial proprieties associated to a high content of phenolic compounds [15] as meat products curing agents could represent an example of circular economy. Nevertheless, more studies are needed to assess the feasibility of frankfurter-type sausage production by replacing sodium and potassium salts of nitrite and nitrate with natural antioxidants, trying both to maintain quality traits and antimicrobial proprieties during storage. The natural extracts were chosen both for their properties and for the link with territory. In fact, the great availability of agricultural byproducts (chestnut, grape seeds, and olive pomace) in Tuscany, where Cinta Senese pigs are reared, was considered.

The present study considers Cinta Senese frankfurter-type sausage, henceforth called sausage and aims to (i) test the use of grape seeds, chestnut, and olive oil as natural extracts to totally replace sodium and potassium salts of nitrite and nitrate; (ii) evaluate the microbiological, physical, and sensorial qualities traits during the storage time (7, 30, 60 days).

## 2. Materials and Methods

### 2.1. Natural Mixtures

The natural antioxidant and antimicrobial extracts used in the present study is based on:grape seeds condensed tannins;hydroxytyrosol, hy-derivatives, and tyrosol from olive oil pomace of *Olea europaea* L.;chestnut hydrolysable tannins.

The extraction of polyphenols from olive oil byproducts is based on a water extraction and membrane separation system. This methodology was instead implemented at an industrial level by using physical technology (PCT/IT/2009/09425529 “Process for producing concentrated and refined active substances from tissues and by-products of *Olea europaea* L. with membrane technologies”) considered sustainable technologies defined by Best Available Technology (BAT) and recognized by the Environmental Protection Agency (EPA) [20,21].

The hydrolysable chestnut tannin extract was obtained following the technology described by Campo et al. [22] (2016), while the extraction of bioactive compounds from the grape seed production was obtained as described by Lucarini et al. [23].

Antioxidant and antiradical capacity and phenolic content of the different extracts were analyzed by HPLC/DAD/MS [19] on three samples of each natural extract and reported in Table 1. Total polyphenolic content of olive pomace extract, grape seed extract and chestnut extract was 38.64 g/L, 822,713 mg/g,, and 16,109 mg/g, respectively. The 1,1-diphenylpicrylhydrazil radical (DPPH) assay on the three components pointed out an antiradical activity (EC50) of 0.196 mg/mg for the olive pomace, 0.147 mg/mg for grape seed extract, 0.085 mg/mg for chestnut extract.

The three extracts were combined to the same amount of hydroxytyrosol and tocopherol (E307) to form a defatted mixture (mix NAT), but their relative amounts are currently under patent and will not be reported in this study.

### 2.2. Sample Manufacturing

Meat used in the trial was derived from 24 Cinta Senese barrows reared outdoors in “Azienda Agricola Savigni” farm (Pistoia, Italy), within the normal running of the farm. Animals were slaughtered at the same age of 15.5 months and at an average live weight of 160 ± 10 kg in a commercial slaughter house as the usual customs of the farmer.

According to the experimental design “2 Lot × 2 Treatment”, the meat was divided in two parts in order to create two lots of products, each including the meat of 12 animals.

For each lot, portions of trimmed pork lean (16 kg) and subcutaneous backfat (4 kg) were used. The meat was chopped into cubes of approximately 3 cm, ground in a commercial food processor, and homogenized adding ice (13.45% of the total recipe) in a cutter (Laska Cutter KU65, Traun, Austria) for 6 min at maximum 10 °C. Following the traditional recipe condiments (salt 1.55%, sucrose 0.25%, black pepper 0.04%) and other seasoning additives (0.47%) were added and therefore the full dough was divided in two parts:one with addition of nitrite and nitrate (1%) as curing agents forming the first treatment (NIT);one with addition of “mix NAT” (1%) to replace nitrite and nitrate and forming the second treatment (NAT).

Each treatment dough was again homogenized with the same quantity of ice in a cutter for 6 min not exceeding 10 °C and then mechanically stuffed (Omet Foodtech, Siena, Italy) into edible collagen casings (Fcase, Myślenice, Poland) (28 mm diameter).

Sausages were steam cooked at 80 °C for 90 min achieving an internal temperature of 72 °C producing about 140 sausages for each treatment. Sausages were immediately chilled with cold water shower, vacuum packed, and stored under refrigeration (4 °C) for 7, 30, and 60 days.

### 2.3. Microbiological, Chemical, and Physical Analysis

Four sausages for each treatment were submitted to microbiological analysis performed at 7, 30, and 60 days of storage time.

Microbiological analyses were carried out in an external accredited laboratory to determine the product safety. The following bacteria were investigated: *Escherichia coli* [24], *Listeria monocytogenes* [25], coagulase positive *Staphylococcus* spp. [26], *Salmonella* spp. [27], total bacterium at 30 °C [28].

Moisture, total protein, and ash contents were determined following AOAC [29] methods. (ii) Total lipid content was analyzed using a modified method of Folch et al. [30] and fatty acid profile of total lipids using the modified technique of Morrison and Smith [31]. Fatty acids (FAs) methyl esters were determined by gas chromatography using a Varian 430 apparatus (Varian Inc., Palo Alto, CA, USA) equipped with a flame ionization detector. FAs separation occurred in a Supelco Omegawax TM 320 capillary column (30-m length; 0.32 mm internal diameter; 0.25 µm film thickness; Supelco, Bellafonte, PA, USA). The individual methyl esters were identified by their retention time using an analytical standard (F.A.M.E. Mix, C8-C22 Supelco 18,920- 1AMP). Response factors based on the internal standard (C19:0) were used for quantification and results were expressed as g/100 g of sample. Six sausages for each treatment were analyzed for chemical composition and physical parameters at the considered storage time (7, 30, and 60 d).

All the physical parameters were assessed on three replications for each of six samples of the two treatments. pH was measured at room temperature (20 °C) using a pH meter Delta Ohm HD 8705 (Delta Ohm S.R.L., Caselle di Selvazzano, Padova, Italy) with a temperature probe TP870 and pH electrode Hamilton double pore. Water activity (aW) was determined by ISO21807:2004 methodology.

Color parameters CIE L* (lightness), a* (redness), and b* (yellowness) were measured immediately after slicing using a Minolta colorimeter CR-200 (Minolta Camera Co., Ltd., Osaka, Japan). Recalibration on white and red plates was performed at the start of each measuring session.

Warner–Bratzler shear force and texture profile analysis (TPA) was performed using a Zwick Roell Z2.5 apparatus (Ulm, Germany texture analyzer) with a 1 kN-load cell at the crosshead speed of 1 mm/s and working at room temperature (22 °C). Tenderness was carried out by Warner–Bratzler shear force applying a tangential cut to a segment of a whole sausage. TPA curve-forces were determined by a 100-mm-diameter compression plate on 10 × 10 ×10 mm slices. Relative to TPA parameters, hardness, springiness, cohesiveness were recorded while chewiness was calculated [32].

### 2.4. Sensory Analysis

Sensory analysis was carried out in an equipped laboratory by nine trained panelists using a quantitative-descriptive analysis method.

Eleven attributes (overall acceptability, color uniformity, lightness, odor, off odor, off flavor, flavor, bitter taste, tannic taste, juiciness, hardness) were evaluated; each attribute was scored in a 100 mm nonstructured line, anchored at the extremities [33]. The list of attributes and their definitions (Appendix A) were presented to selected panelists underwent an introductory session, where the testing procedures and the chosen sensory traits were discussed using two types of comparable commercial products. During three sessions, panelists evaluated a total of eight sausages (2 lot × 2 treatments × 2 samples) identified by an alphanumerical code. The sausages were divided in 5 mm thick pieces and two pieces of each sample were randomly served to judges at room temperature (20 °C).

The sausages’ sensory characteristics were evaluated exclusively at the storage time of 60 days.

### 2.5. Statistical Analysis

Data were analyzed by MIXED Procedure of SAS Software (SAS, 2007) according to the following model (1):Y_ijkl_ = µ + T_i_ + S_ij_ + L_k_+ D_l_ + (TxD)_il_ + ε_ijkl,_(1)
where µ was the mean, T was the i^th^ treatment, L was the k^th^ production lot, D was the lth day of storage, (TxD) was the interaction between treatment and day of storage, S was the effect of the sample within treatment and it is equal to the covariance between repeated measurements within samples; ε was the random error with mean 0 and variance σ^2^, i.e., the variance between measurement within samples.

For the sensory data, GLM procedure including in the model the effect of panelist (P) was used as shown in the following Equation (2):Y_ijkl_ = µ + T_i_ + S_ij_ + L_J_ + P_k_ + ε_ijkl_.(2)

Level of significance was started at *p* < 0.05. Tukey test was used to statistically test the differences between the least squares means.

## 3. Results

### 3.1. Microbiological Characterization

Table 2 shows the microbiological population in sausages at 7, 30, and 60 days. In all samples considered, the major foodborne pathogens (*Escherichia coli*, *Listeria monocytogenes*, *Staphilococcus* spp., and *Salmonella* spp.) were absent or below the legal threshold limit (Reg CE/2073/05). The total microbial population showed the same starting point but, as expected, a slight increase was observed during the whole storage period for both sausage groups. Between treatments, differences occurred from 30 days onwards.

### 3.2. Chemical Traits

Regarding chemical composition during storage (Table 3), data were reported as mean value between treatments considering that the same raw materials were used for all treatments and no difference appeared between the experimental groups. Moisture content showed a decreasing trend: differences were significant from 30 to 60 days and, consequently, protein, fat, and ash content appeared slightly increased at 60 days.

Regarding fatty acids profile (Table 4), the main fraction was represented by MUFA, followed by the SFA and PUFA in both types of sausage. In all samples, the predominant individual fatty acid was oleic acid (C18:1) followed by palmitic (C16:0), linoleic (C18:2n6), stearic (C18:0). PUFA/SFA value, useful to determine the nutritional quality of the food lipid fraction, was higher than 0.42 in both sausage types.

Relative to the storage times, the main categories of fatty acids did not show significative differences during the considered period. Some single fatty acid significantly changed their content at 60 days in NIT sausages (i.e., palmitic and palmitoleic acids), while in NAT sausages stearic acid decreased from 7 to 30 days.

NAT sausages obtained a slightly lower content of MUFA and a higher content of polyunsaturated, while for SFA no differences were shown. These differences reflected the behavior of the single fatty acids such as palmitoleic, oleic, and linoleic. Indeed, already at 7 days, PUFA content in NAT sausages was higher than MUFA ones. Consequently, the PUFA/SFA ratio showed significant differences between treatments: NIT sausages had always the lowest values.

### 3.3. Physical Traits

The pH showed a range between 5.39 and 6.19 and decreased from 7 to 60 days in both sausage types. Significant differences among treatments were found, with a slightly lower values observed for NAT groups respect to NIT (Table 5).

Water activity (aW), showed a range between 0.977 to 0.988 and no differences were found both between treatments and storage times.

Regarding color data, lightness remained constant during the storage period in NAT sausages while decreased in NIT groups from 30 to 60 days. Redness progressively increased in NAT sausages as storage time increased, while in NIT sausages remained stable. Yellowness slightly decreased its value from 7 to 30 days in NAT, while in NIT the decline was from 30 to 60 days.

All color attributes were affected by treatment: L* showed significantly greater values in NIT samples than NAT ones excluding lightness at 60 days; a* was nearly twice as red in NIT, while b* was significantly higher in NAT at 7 and 30 days compared to the control. Regarding the tenderness performed by Warner–Bratzler (WB) there were no significant differences in storage times, whereas there were between treatments with NIT samples always less tender than NAT.

Considering texture profile analysis (TPA), hardness did not change over time in NAT sausages, while in NIT products showed a significant increase from 7 to 60 days. Storage time affected the cohesiveness in NAT sausages showing an increase from 30 to 60 days, while springiness and chewiness did not change over time in both sausage types.

Except for springiness, all TPA parameters are affected by treatment, being higher in NIT samples than NAT.

### 3.4. Sensory Characterization

The results of sensory evaluation (Figure 1) indicated that the major part of attributes was influenced (*p* < 0.05) by replacement of synthetic curing agents with natural extracts. The sensorial characteristics of the products showed that the color of NAT sausages was less uniform color and was two times less lightness compared to NIT ones. Differences between sausage types in odor and flavor attributes occurred with more intensity; NAT sausages also recorded higher average scores for off odor and off flavor, identified as bitter and tannic taste by panelists (data not reported).

According to TPA instrumental results, NAT sausages were evaluated more tender by panelists with respect to NIT, while no difference among samples were observed in juiciness.

The NIT sausages obtained a score evidently higher than NAT sausages.

## 4. Discussion

Sausages were safe for human consumption throughout the entire storage period and no visual spoilage was evident. Even if total microorganisms increase during the time, they did not cause alteration harmful for human consumption. Additionally, Rannucci et al. [34] observed an increase of total microorganisms during storage and in particular from 0 until 18 days in frankfurter sausages manufactured using an ancient recipe (pork meat from shoulder) but, even in this case, no pathogens were developed. Indeed, the antibacterial effect of some components included in NAT MIX was already documented, such as the effect of grape seed tannin against *Escherichia coli* [35] and of hydrolysable tannins against *Salmonella* and *Stafylococcus* [36].

The natural mixture seemed to be effective in terms of food safety showing lower values of total microbial count, probably because phenolic compounds chelate some metal ions of meat required for microbial growth [37]. A higher microbial growth could have played a role in degradation of different structural proteins, and, thus, tannins of natural mixtures, with their antioxidant and antimicrobial activity [38], prevented protein degradation of NAT products. The absence of main foodborne pathogens in natural curing agent sausages suggested that they provide the same antibacterial effect of synthetic curing agents. Hence, antimicrobial properties of natural curing agents could intervene in the delay of microbial spoilage as also suggested by Maqsood et al. [38] and Fasolato et al. [39].

However, further specified studies would be required to define the effectiveness of antimicrobial activity on products.

Water activity, often related with packaging and the sausages being a wet product, was always quite high, and this did not contribute to avoiding microbial growth. Ranucci et al. [34] obtained an aW value around 0.97 defining a high aW value that could promote the growth of such microbial populations even if samples were stored under vacuum refrigeration conditions. Alirezalu et al. [18] in frankfurter-type sausages with added nisin, polylysine, chitosan and natural extracts found similar aW values from 0.97 to 0.98 and, as in our case, the natural extract treatments did not affect this parameter.

Chemical composition of sausages suggested that they were less moist and more fat than similar products and even if stored under vacuum condition, they had lost part of the aqueous component up to 30 days. This could be probably due to Cinta Senese meat that had higher content of intramuscular fat than commercial breeds usually employed to produce sausages, as reported by Franci et al. [40]. Furthermore, the higher amount of fat in sausage products could negatively affect consumer opinion because a high daily fat intake is dangerous for human health, being a factor linked to the development of obesity, coronary heart disease, and arterial hypertension.

The fatty acid composition of sausages was characterized by lower level of SFA and higher levels of MUFA and PUFA compared to data reported in literature for pork frankfurter sausages [18,34,41]. As in this study, Škrlep et al. [42], in dry fermented sausages, recorded a PUFA amount of about one third of MUFA content and in parallel a low SFA content. This proportion of fatty acid groups can be associated to the outdoor rearing system, in particular to the availability of green feed. Additionally, lower fat saturation may be linked to higher physical activity of the pigs used in this study, as suggested by Daza et al. [43] who showed lower SFA in the backfat and higher PUFA in muscle of pigs reared outdoor than sedentary ones fed with the same diet.

The predominant individual fatty acids (oleic, palmitic, stearic, linoleic), typical of pork meat, were the same that occurred in the studies of Alirezalu et al. [18] and Fonseca et al. [44] in frankfurter and traditional Spanish sausages.

PUFA/SFA value was higher in both sausage types with respect to the threshold value indicated for healthiness of meat products (>0.40) reported by Wood et al. and with respect to other frankfurter sausages which reported values from 0.31 to 0.37 [16] and from 0.35 to 0.40 [18]. Additionally, Ranucci et al. [34], applying an ancient recipe for sausage preparation, showed a PUFA/SFA ratio above 0.40 associating these results with the lower value of saturated fatty acids.

The absence of differences in fatty acid composition and PUFA/SFA ratio during the storage time would seem to suggest that if process storage is conducted following the good practice, fatty acids do not change their profile. Furthermore, the lack of change of acidic profile during storage time, in particular of PUFA, denoted the positive influence of NAT and NIT curing agents on lipid oxidation. Both this lack of change together with the non-proliferation of pathogenic microorganisms are positive results in terms of product shelf-life.

Regarding the treatments, if the PUFA are considered as an indicator of meat oxidative status as proposed by Pateiro et al. [15], the slightly higher content in NAT already present at day 7 suggested that the natural curing agents employed had been less effective in the lipid oxidation control during the first part of the manufacturing process. Nevertheless, the addition of natural extract ingredients had only a “little impact” on fatty acid composition according to Alirezalu et al. [18].

The fatty acids results of this study seem to be linked to the use of Cinta Senese meat and consequently both to the genetic characteristic of the breed and to the specific outdoor rearing system, that included almost natural resources feeding. In fact, the fatty acid profile of local pigs is characterized to a greater deposition of MUFA, mainly oleic acid, than improved pigs whose fat contains higher quantities of saturated fatty acids. The highest level of monounsaturated fatty acids in native breeds is a consequence of differences in de novo lipid synthesis and their capacity to deposit monounsaturated fatty acids increases with age [45].

Regarding the physical traits, the pH showed values within the range obtained by other researches on frankfurter products [18,46]. The pH decline during the refrigerated vacuum package storage in both sausage types can be associated with meat intrinsic properties and/or microbial fermentation and in particular with the growth of lactic acid bacteria as suggested in study of Viuda-Martos et al. [47]. The pH differences recorded between treatments can be linked to the presence of *Lactobacillus* as observed by Hospital et al. [48] in low or nitrites-free sausages. This fact could be the cause of pH decline in NAT sausages. Nevertheless, the microbiota populations were not assessed in the present experiment and it is difficult to make any further conclusions. Regarding the color, higher redness of NIT sausages and its major stability was probably linked to the role of nitrites in nitrosyl-hemochrome formation which contributed both to characteristic red pigment of the commercial meat product and to its stability during storage times [4,7]. In addition, Özvur et al. [49] observed lower values of red color in products with added grape seed extract and they attributed the differences to natural color and emulsion properties of treatment. Ranucci et al. [34] obtained nitrate free frankfurter sausages with color parameter similar to this study (low a* and higher L* values) and they linked the result to the muscle and fat types that were different in their recipe.

Furthermore, in NAT sausages the lower color intensity could be associated to the oxidative reactions of lipids linked to the formation of yellow colored polymers [50] while the lower redness could be linked to the oxidation of muscle pigment myoglobin probably depending on lower pH [51].

It seems to be accepted that the main factors affecting the characteristic cured meat color is the nitrosylmyoglobin formation through the reaction of nitrite with myoglobin in the presence of endogenous or added reductants [52]. In absence of nitrites, natural extracts did not play this role and the sausages appear different.

The effect of treatments on sausages tenderness suggested that the NAT group were less hard than NIT, irrespective of the used method. The different result between Warner–Bratzler shear force and texture profile analysis could be associated to the different force that acts on the sample during the analysis. Novakovi et al. [32] in a review where they compared the two methods, suggested that the WB was less accurate because it has the highest coefficient of variability. Natural mixture also affected texture profile attributes—hardness, cohesiveness, and chewiness. Those results can be attributable basically to the differences in pH, which, when declining, causes the aggregation of myofibrillar proteins [53]. The higher pH of NIT samples could have led to the formation of a gel with myosin and entrapped water, leading to a stronger structure of sausages. In accordance with our results, Pateiro et al. [15] also reported that the hardness values significantly decreased with the addition of antioxidants. However, to the best of our knowledge, few data are available about frankfurter-type sausages with natural extract as single curing agents and the great variability of these traditional products makes comparisons difficult.

The sensory evaluation of sausages confirmed the differences in color and tenderness between products. Lower color uniformity and less lightness could be linked to the level of oxidation but also to the effect of nitrite on these attributes. Furthermore, in line with the results obtained from instrumental texture evaluation, NAT sausages were evaluated more tender by panelists. Odor and flavor were more intense in NAT sausages, even if these attributes were associated to off odor, off flavor, and other sensations. These results could be linked both to the remaining taste of the primary natural mixture constituents and to the effect of synthetic curing agents on products. In fact, Braghieri et al. [33] working on sensory evaluation of sausages suggested that, flavor being affected by several factors, differences in odor and flavor between products may be due to the inhibition of most activities as endogenous enzyme activities, microbial activities, and autoxidation in nitrate products. Regarding the overall acceptability, NIT samples obtained an evidently higher score than NAT sausages. Panelists considered synthetic curing agent sausages acceptable while the overall acceptability of natural extract product needs to be improved. This result was probably affected by many factors such as off flavor and off odor of NAT sausages but also by the consumers’ habit of traditional taste and texture of synthetic curing agents’ sausages. According to Alirezalu et al. [18] panelists preferred more compact sausages attributing higher scores to sausages with high hardness values. Similarly, Ribas-Agustí et al. [54] working on dry fermented sausages reported that panelists discarded the products with grape seed extract, because judged to be abnormal with respect to control samples. In addition, Özvural [55] observed a decrease in overall acceptability of frankfurters with grape seed extract added.

Sensory characteristics together with safety traits represent the primary attributes for consumers quality evaluation, followed by ethical and nutritional factors [33]. In terms of sensory properties, taste is the most important factor followed by color and texture. If these factors effectively influence the consumer intention to purchase [51], the results of sensory analysis of this study suggested that these attributes could be improved by a revaluation of the recipe of natural extracts sausages.

## 5. Conclusions

Nitrite/nitrate and natural frankfurter-type sausage was safety for the human consumption, as none of the main foodborne pathogens were found during the storage time considered.

Both types of sausages had a higher content of fat probably associated with the higher intramuscular fat of the Cinta Senese breed. Considering that the higher amount of daily lipid intake is considered negative for human health, manufacturers could re-evaluate the recipe and decrease the amount of fat.

The fatty acid composition showed that NAT treatment would seem slightly less efficient on lipid oxidation during the manufacturing, while in all storage times both treatments did not implicate remarkable impacts. Some differences in color suggested that further research is needed to improve color stability of natural extract sausages both during processing and storage. The replacements of natural extract as curing agents changed the perception of some sensorial properties overall in terms of taste and odor, recognized as “off” attributes and, as a whole, had a negative effect on the sensorial evaluation of NAT sausages.

Finally, the results so far indicated that natural extracts are an alternative to synthetic additives in Cinta Senese sausages even if some attributes could be improved.

The implementation of a new product using second-choice portion meat of a local breed and agricultural by products as curing agents could be interesting in terms both of sustainability and valorization of the territory and of Cinta Senese products.

## Figures and Tables

**Figure 1 animals-10-00014-f001:**
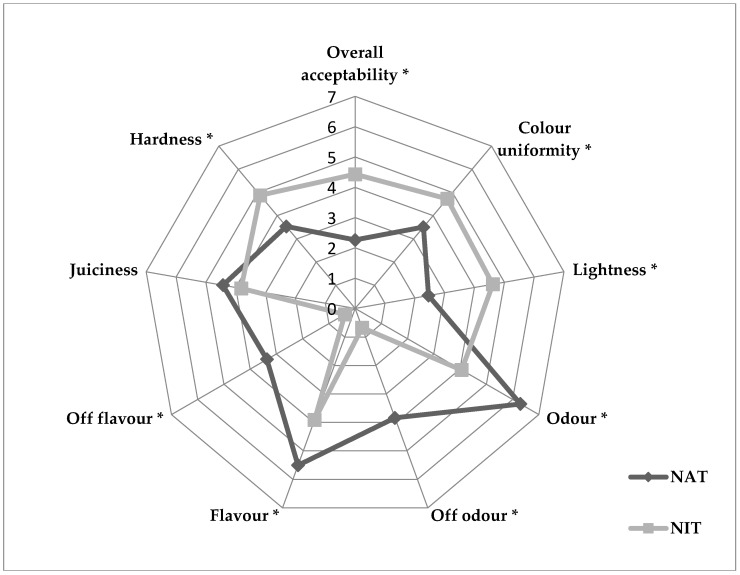
Sensorial traits of Cinta Senese sausages manufactured with natural extracts (NAT) as replacement of sodium nitrite (NIT), “*” indicates significant differences (*p* < 0.05).

**Table 1 animals-10-00014-t001:** Phenolic profile of the extracts combined in the mixture (natural mixture—NAT) used in the trial.

Olive Oil Pomace	g/L	Grape Seed	mg/g	Chestnut Tannin	mg/g
OH-tyrosol	11.65	Catechin	11.07	Vescalin	9.34
OH-tyrosol derivatives	5.13	Epicatechin	13.62	Castalin	9.00
Tyrosol	16.02	Catechin dimers	10.21	Pedunculagin	3.88
Verbascoside	5.84	Catechin trimers	6.92	Galloil glucose derivatives	42.59
		Epicatechin gallate derivatives	726.02	Gallic Acid	18.50
		Tetramers	54.88	Roburin D	10.51
				Vescalagin	32.15
				Castalagin	31.03
				Ellagic Acid	4.08

**Table 2 animals-10-00014-t002:** Microbiological analysis on Cinta Senese sausages manufactured with natural antioxidant (NAT) in replacement of sodium nitrite (NIT).

Parameter	Treatment	Days of Storage			T × D
		7	30	60	
***Escherichia coli***	NAT	<1.0	<1.0	<1.0	n.s.
	NIT	<1.0	<1.0	<1.0	n.s.
***Listeria monocytogenes***	NAT	−	−	−	n.s.
	NIT	−	−	−	n.s.
**Coagulase positive *Staphilococcus* spp.**	NAT	<1.0	<1.0	<1.0	n.s.
	NIT	<1.0	<1.0	<1.0	n.s.
***Salmonella* spp.**	NAT	−	−	−	n.s.
	NIT	−	−	−	n.s.
**Total microbial count (30 °C)**	NAT	7.04 ^a^	8.14 ^b^^,A^	10.20 ^c,A^	n.s.
	NIT	7.04 ^a^	8.32 ^b,B^	10.32 ^c,B^	n.s.

Results are expressed as log 10 colony forming units (CFU) per g of sausage; the symbol “−” indicates that the organism was not present. NAT = sausage with natural extracts, NIT = sausage with nitrite and nitrate as curing agents, ^a,b^^,c^ = different letters in the same row indicate significant differences (*p* < 0.05), ^A,B^ = different letters in the same column indicate significant differences (*p* < 0.05), T × D = interaction treatments × days of storage, n.s. = not significant.

**Table 3 animals-10-00014-t003:** Chemical composition of sausage during storage times (%).

Parameter	Days of Storage			T × D
	7	30	60	
**Moisture**	52.04 ^b^	51.71 ^b^	50.07 ^a^	n.s.
**Protein**	15.26 ^a^	15.36 ^a^	15.97 ^b^	n.s.
**Lipid**	30.26 ^a^	30.51 ^a^	31.42 ^b^	n.s.
**Ash**	2.44 ^a^	2.42 ^a^	2.54 ^b^	n.s.

^a,b^ = Different letters in the same row indicate significant differences (*p* < 0.05). T × D = interaction treatments × days of storage, n.s. = not significant.

**Table 4 animals-10-00014-t004:** Fatty acid profile during storage times in natural (NAT) and nitrite/nitrate (NIT) sausages.

Parameter	Treatment	Days of Storage			T × D
		7	30	60	
**C16:0**	**NAT**	23.17	23.09	23.19 ^A^	n.s.
	NIT	23.31 ^a^	23.27 ^a^	23.61 ^b,B^	n.s.
**C16:1**	NAT	2.63 ^A^	2.64 ^A^	2.64 ^A^	n.s.
	NIT	2.70 ^a,B^	2.70 ^a,B^	2.75 ^b,B^	n.s.
**C18:0**	NAT	12.17 ^b^	11.96 ^a^	12.06 ^a,b^	n.s.
	NIT	12.16	12.09	12.07	n.s.
**C18:1**	NAT	41.85 ^A^	42.16	41.94	n.s.
	NIT	42.62 ^B^	42.40	42.31	n.s.
**C18:2n6**	NAT	14.44 ^B^	14.45 ^B^	14.44 ^B^	n.s.
	NIT	13.58 ^A^	13.80 ^A^	13.65 ^A^	n.s.
**C18:3n3**	NAT	0.94 ^B^	0.94 ^B^	0.94 ^B^	n.s.
	NIT	0.88 ^A^	0.90 ^A^	0.89 ^A^	n.s.
**SFA**	NAT	37.38	37.03	37.29	n.s.
	NIT	37.51	37.45	37.73	n.s.
**MUFA**	NAT	45.70 ^A^	46.04 ^A^	45.79 ^A^	n.s.
	NIT	46.59 ^B^	46.35 ^B^	46.30 ^B^	n.s.
**PUFA**	NAT	16.90 ^B^	16.92 ^B^	16.91 ^B^	n.s.
	NIT	15.90 ^A^	16.20 ^A^	16.00 ^A^	n.s.
**n3PUFA**	NAT	1.20 ^B^	1.21 ^B^	1.21 ^B^	n.s.
	NIT	1.14 ^A^	1.17 ^A^	1.14 ^A^	n.s.
**n6PUFA**	NAT	15.67 ^B^	15.68 ^B^	15.67 ^B^	n.s.
	NIT	14.72 ^A^	14.98 ^A^	14.80 ^A^	n.s.
**PUFA/SFA**	NAT	0.45 ^B^	0.46 ^B^	0.45 ^B^	n.s.
	NIT	0.42 ^A^	0.43 ^A^	0.42 ^A^	n.s.

NAT = sausages with natural extracts, NIT = sausages with nitrite and nitrate as curing agents, ^a,b^ = different letters in the same row indicate significant differences (*p* < 0.05), ^A,B^ = different letters in the same column indicate significant differences (*p* < 0.05), T × D = interaction treatments × days of storage, n.s. = not significant.

**Table 5 animals-10-00014-t005:** Physical traits during storage times in natural (NAT) and nitrite/nitrate (NIT) sausages.

Parameter	Treatment	Days of Storage			T × D
		7	30	60	
**pH**	NAT	6.07 ^c,A^	5.62 ^b,A^	5.39 ^a,A^	n.s.
	NIT	6.19 ^c,B^	5.74 ^b,B^	5.53 ^a,B^	n.s.
**Water activity (aW)**	NAT	0.977	0.977	0.978	n.s.
	NIT	0.989	0.989	0.988	n.s.
**L***	NAT	66.84 ^A^	66.84 ^A^	67.00 ^A^	n.s.
	NIT	69.62 ^b,B^	69.97 ^b,B^	67.89 ^a,A^	n.s.
**a***	NAT	8.22 ^a,A^	9.77 ^bA^	11.22 ^c,A^	*
	NIT	17.02 ^B^	17.03 ^B^	17.27 ^B^	
**b***	NAT	12.43 ^a,B^	11.55 ^b,B^	11.61 ^b^	*
	NIT	11.17 ^a,A^	11.13 ^a,A^	11.46 ^b^	
**Warner–Bratzler shear force (kg)**	NAT	29.48 ^A^	30.54 ^A^	31.22 ^A^	n.s.
	NIT	37.78 ^B^	39.86 ^B^	39.55 ^B^	n.s.
**TPA**					
**hardness (N)**	NAT	71.54 ^A^	70.21 ^A^	74.76 ^A^	*
	NIT	78.02 ^a,B^	86.20 ^a,b,B^	93.28 ^b,B^	
**Cohesiveness**	NAT	0.55 ^a,b,A^	0.51 ^a,A^	0.59 ^b^	*
	NIT	0.67 ^B^	0.66 ^B^	0.62	
**Springiness (mm)**	NAT	7.59	7.69	7.33	n.s.
	NIT	7.65	7.69	7.67	n.s.
**Chewiness (Nx mm)**	NAT	300.03 ^A^	284.77 ^A^	300.05 ^A^	n.s.
	NIT	398.97 ^B^	434.53 ^B^	441.85 ^B^	n.s.

NAT = sausages with natural extracts, NIT = sausages with nitrite and nitrate as curing agents, ^a,b^ = different letters in the same row indicate significant differences (*p* < 0.05), ^A,B^ = different letters in the same column indicate significant differences (*p* < 0.05), T × D = interaction treatments × days of storage, n.s. = not significant, * = significant (*p* < 0.05).

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
