# Peer review of "Effect of Replacement of Synthetic vs. Natural Curing Agents on Quality Characteristics of Cinta Senese Frankfurter-Type Sausage"

_animals, 2019, doi:10.3390/ani10010014_

Round 1

Reviewer 1 Report

An exhaustive paper, this one, able to catch the reader's attention up to its very end, though its style and property of term selection might be improved.

Language: several distraction errors occurred (such as at line 23, line 202 (are you kidding: slices 5-cm thick?) or when it comes to the agreement between subject and verb); choice of time is not always accurate. The term "Organoleptic" should be banned, to be replaced by "Sensory" and/or "Sensorial"; besides, what does it mean that "the natural extract determined the same antibacterial powerful of NIT" (line 315-316)? Definitely, the paper would gain a great deal thank to a thorough linguistic revision.

Sensory analysis methods: I do believe that a more detailed description of what was done would do a great job for this paper. At the same time, alas, it seems to me that the selected attributes, as listed at lines 197-198, were quite generic, so that they should have been painstakingly verbally predefined, in order to prevent any sort of blunder. I am pretty sure that these definitions were outlined (otherwise it shoul have been nearly impossible to work) and therefore would you be so kind as to include in the paper this list of terms, with their suitable definition and frame of reference?

References: Campo et al. (2016) has not been listed at the end of the paper; this apply also to the UNI EN ISO norms (line 165-168)

Tables: an unusual use of the lettering caught my attention. For instance, have a look at table 3, where for Moisture the sequence is a, a, b; the same applied to Protein, Lipid and Ash, whereas it should have been the reverse. Table 5 sounds strange as to pH data, when compared to what has been written at line 262. Besides, whenever a table must be splitted, was it not necessary to repeat its title?

Author Response

Dear Editor and Reviewers,

we considered and addressed the suggestions proposed. When possible, punctual corrections were made in the text and the changes were highlighted in yellow. Supplementary material has been added.

Title has been changed as suggested by Reviewer 2.

References have been slightly modified as suggested by reviewer 2 and consequently renumbered.

Linguistic improvements have been applied throughout the text.Best regards, Silvia Parrini

Reviewer 1

Suggestions for Authors: An exhaustive paper, this one, able to catch the reader's attention up to its very end, though its style and property of term selection might be improved.

Comments to the Author

Language: several distraction errors occurred (such as at line 23, line 202 (are you kidding: slices 5-cm thick?) or when it comes to the agreement between subject and verb); choice of time is not always accurate.

AU: Done, text has been improved. Regard the samples used in the sensorial evaluation there was a mistake in the measure unit (thick pieces of 5 mm), we corrected the text (L210-212)

Comments to the Author

The term "Organoleptic" should be banned, to be replaced by "Sensory" and/or "Sensorial";

AU: Organoleptic is replaced by the correct term (L 25 and 62)

Comments to the Author

What does it mean that "the natural extract determined the same antibacterial powerful of NIT" (line 315-316)?  

AU: Text (L 329-330) have been improved.

Comments to the Author

Definitely, the paper would gain a great deal thank to a thorough linguistic revision.

AU: Linguistic improvements have been applied

Comments to the Author

Sensory analysis methods: I do believe that a more detailed description of what was done would do a great job for this paper. At the same time, alas, it seems to me that the selected attributes, as listed at lines 197-198, were quite generic, so that they should have been painstakingly verbally predefined, in order to prevent any sort of blunder. I am pretty sure that these definitions were outlined (otherwise it should have been nearly impossible to work) and therefore would you be so kind as to include in the paper this list of terms, with their suitable definition and frame of reference?

AU: Sensorial attributes were previous predefined but not included in the paper. As you suggest, additional information has been added as supplementary materials.

Comments to the Author

References: Campo et al. (2016) has not been listed at the end of the paper; this apply also to the UNI EN ISO norms (line 174-177)

AU: References have been added and consequently renumbered.

Comments to the Author

Tables: an unusual use of the lettering caught my attention. For instance, have a look at table 3, where for Moisture the sequence is a, a, b; the same applied to Protein, Lipid and Ash, whereas it should have been the reverse.

AU: All tables have been corrected.

Comments to the Author

Table 5 sounds strange as to pH data, when compared to what has been written at line 262. Besides, whenever a table must be splitted, was it not necessary to repeat its title?

AU: Ok, text has been improved (L 273-274) and repeated titles have been removed.

Reviewer 2 Report

This study aimed to evaluate effect of natural additives such as olive oil by-products, grape seed, chestnut tannin on quality characteristics and alternatives for nitrate/nitrite of frankfurter type sausage.

The idea is interesting, however, logical explanation is not enough to explain implication of the study. In some parts, anecdotal words were used instead of scientific terms. Authors should correct the entire manuscript in this point of view.

There are several points to be corrected.

-Authors should rephrase the title to show main topic of the study. Using alternative for nitrite and nitrate.

-In abstract, line 30-31, provide specific kinds of each pathogens.

-Materials and methods are not enough to show golden methods for the study.

-Olive oil, grape seed, chestnut tannin were used in this study. Why authors chose those specific natural compounds? Need explanation.

-Need IRB approval number for animal experiment.

- Microbial analysis should explained with TBARS, VBN measurements.

-Footnotes in the table should be amended. For example, I cannot find TxD interaction data.

-Overall acceptability less than 2.5 cannot accepted for the quality of the sausage. Need more profound discussion for this.

Author Response

Dear Editor and Reviewers,

we considered and addressed the suggestions proposed. When possible, punctual corrections were made in the text and the changes were highlighted in yellow. Supplementary material has been added.

Title has been changed as suggested by Reviewer 2.

References have been slightly modified as suggested by reviewer 2 and consequently renumbered.

Linguistic improvements have been applied throughout the text.

Best regards,

Silvia Parrini

Comments to the Author

In some parts, anecdotal words were used instead of scientific terms. Authors should correct the entire manuscript in this point of view.

AU: we did our best to improve the text but it would have been more useful if the reviewer had provided more details on anecdotal words

Comments to the Author

-Authors should rephrase the title to show main topic of the study. Using alternative for nitrite and nitrate.

AU: done (L2-6)

Comments to the Author

-In abstract, line 30-31, provide specific kinds of each pathogens.

AU: done (L34-35)

Comments to the Author

-Materials and methods are not enough to show golden methods for the study.

AU: We apologize but we really didn’t get the point raised up by the reviewer. We reported the information normally used in the scientific paper for this type of methodologies. Anyway, we will be grateful if the reviewer could provide additional information at this respect.

Comments to the Author

- Olive oil, grape seed, chestnut tannin were used in this study. Why authors chose those specific natural compounds? Need explanation.

AU: Olive oil, grape seed, chestnut tannin were used both for their proprieties and for the link with the territory. We improved the text (L105-114)

Comments to the Author

-Need IRB approval number for animal experiment.

AU: The trial was carried out on meat deriving from Cinta Senese pigs and not on the live animals. In order to avoid confusion, text has been changed (L147-154). Therefore, an ethical statement regarding the use of live animal and their welfare is not necessary as suggested by Editor.

Comments to the Author

- Microbial analysis should explained with TBARS, VBN measurements.

AU: We agree with the referee that TBARS and VNB should better explain the microbial presence, but this aspect has been covered only as an additional part and it doesn't represent the main focus of the paper. We thus decided to did not perform these analyses. 

Comments to the Author

-Footnotes in the table should be amended. For example, I cannot find TxD interaction data.

AU: Tables (2,3,5) have been improved: statistical significance of TxD interaction was reported.

Comments to the Author

-Overall acceptability less than 2.5 cannot accepted for the quality of the sausage. Need more profound discussion for this.

AU: Text has been improved (L418-421).

Round 2

Reviewer 2 Report

This manuscript improved significantly according to reviewer's suggstion.

However, there are stll uncertain answers for the reviewer's comments.

Comments to the author 

- Microbial analysis should explained with TBARS, VBN measurements.

AU: We agree with the referee that TBARS and VNB should better explain the microbial presence, but this aspect has been covered only as an additional part and it doesn't represent the main focus of the paper. We thus decided to did not perform these analyses.

I think the relations between microbes and lipid or protein oxidation should be provided in discussion part. Because, as you said, microbiological analysis was done to show product’s safety. You need discussion. Every data represent important results. If those data did not represents the main focus why did you provided in the manuscript?

-Materials and methods are not enough to show golden methods for the study.

AU: We apologize but we really didn’t get the point raised up by the reviewer. We reported the information normally used in the scientific paper for this type of methodologies. Anyway, we will be grateful if the reviewer could provide additional information at this respect.

-For example, the meat you used for the sausage need specific muscle name (m Longissimus dorsi?) and portion (loin? Round? Leg?). Where is the slaughter house? Was it commercial slaughter house? or just done it in the farm?

-Did you buy the meat in local market?

-To determine fatty acid composition, reference for the methods should be provided.

-Diameter of the sausage was 28 mm and the sample for the TPA test was 1 cm*1 cm*1 cm. Why did you use the sample only inside of the sausage? Need some explanation.

-To determine overall acceptability for sensory evaluation, you cannot train panels.

-Table 3. Authors should amend the footnotes.

For example,

a-b Different letters of the mean value in the same row differ significantly (p<0.05).

TⅹD = interaction treatments ⅹ days of storage, n.s= not significant

-Table 5. What is the meaning of * in the table 5? Is it statistical significance at p<0.05?

Author Response

Dear Reviewer,

we considered and addressed the suggestions proposed. When possible, punctual corrections were made in the text and the changes were highlighted in green.

References have been added and consequently renumbered.

 Best regards,

Silvia Parrini
